# A Boolean Task Algebra for Reinforcement Learning

## Abstract

We propose a framework for defining a Boolean algebra over the space of tasks. This allows us to formulate new tasks in terms of the negation, disjunction and conjunction of a set of base tasks. We then show that by learning goal-oriented value functions and restricting the transition dynamics of the tasks, an agent can solve these new tasks with no further learning. We prove that by composing these value functions in specific ways, we immediately recover the optimal policies for all tasks expressible under the Boolean algebra. We verify our approach in two domains—including a high-dimensional video game environment requiring function approximation—where an agent first learns a set of base skills, and then composes them to solve a super-exponential number of new tasks.

## 1 Introduction

Reinforcement learning (RL) has achieved recent success in a number of difficult, high-dimensional environments (Mnih et al., 2015; Levine et al., 2016; Lillicrap et al., 2016; Silver et al., 2017). However, these methods generally require millions of samples from the environment to learn optimal behaviours, limiting their real-world applicability. A major challenge is thus in designing sample-efficient agents that can transfer their existing knowledge to solve new tasks quickly. This is particularly important for agents in a multitask or lifelong setting, since learning to solve complex tasks from scratch is typically infeasible.

One approach to transfer is *composition* (Todorov, 2009), which allows an agent to leverage existing skills to build complex, novel behaviours. These newly-formed skills can then be used to solve or speed up learning in a new task. In this work, we focus on concurrent composition, where existing base skills are combined to produce new skills (Todorov, 2009; Saxe et al., 2017; Haarnoja et al., 2018; Van Niekerk et al., 2019; Hunt et al., 2019; Peng et al., 2019). This differs from other forms of composition, such as options (Sutton et al., 1999) and hierarchical RL (Bacon et al., 2017), where actions and skills are chained in a temporal sequence.

In this work, we define a Boolean algebra over the space of tasks and optimal value functions. This extends previous composition results to encompass all Boolean operators: conjunction, disjunction, and negation. We then prove that there exists a homomorphism between the task and value function algebras. Given a set of base tasks that have been previously solved by the agent, any new task written as a Boolean expression can immediately be solved without further learning, resulting in a zero-shot super-exponential explosion in the agent's abilities.

We illustrate our approach in a simple domain, where an agent first learns to reach a number of rooms, after which it can then optimally solve any task expressible in the Boolean algebra. We then demonstrate composition in high-dimensional video game environments, where an agent first learns to collect different objects, and then compose these abilities to solve complex tasks immediately. Our results show that, even when function approximation is required, an agent can leverage its existing skills to solve new tasks without further learning.

## 2 Preliminaries

We consider tasks modelled by Markov Decision Processes (MDPs). An MDP is defined by the tuple $(\mathcal{S}, \mathcal{A}, \rho, r)$, where (i) $\mathcal{S}$ is the state space, (ii) $\mathcal{A}$ is the action space, (iii) $\rho$ is a Markov transition

kernel $(s, a) \mapsto \rho_{(s,a)}$ from $\mathcal{S} \times \mathcal{A}$ to $\mathcal{S}$, and (iv) $r$ is the real-valued reward function bounded by $[r_{\text{MIN}}, r_{\text{MAX}}]$. In this work, we focus on stochastic shortest path problems (Bertsekas & Tsitsiklis, 1991), which model tasks in which an agent must reach some goal. We therefore consider the class of undiscounted MDPs with an absorbing set $\mathcal{G} \subseteq \mathcal{S}$.

The goal of the agent is to compute a Markov policy $\pi$ from $\mathcal{S}$ to $\mathcal{A}$ that optimally solves a given task. A given policy $\pi$ is characterised by a value function $V^\pi(s) = \mathbb{E}_\pi \left[ \sum_{t=0}^\infty r(s_t, a_t) \right]$, specifying the expected return obtained under $\pi$ starting from state $s$.[1] The *optimal* policy $\pi^*$ is the policy that obtains the greatest expected return at each state: $V^{\pi^*}(s) = V^*(s) = \max_\pi V^\pi(s)$ for all $s$ in $\mathcal{S}$. A related quantity is the $Q$-value function, $Q^\pi(s, a)$, which defines the expected return obtained by executing $a$ from $s$, and thereafter following $\pi$. Similarly, the optimal $Q$-value function is given by $Q^*(s, a) = \max_\pi Q^\pi(s, a)$ for all $s$ in $\mathcal{S}$ and $a$ in $\mathcal{A}$. Finally, we denote a *proper policy* to be a policy that is guaranteed to eventually reach the absorbing set $\mathcal{G}$ (James & Collins, 2006; Van Niekerk et al., 2019). We assume the value functions for improper policies—those that never reach absorbing states—are unbounded below.

## 3    Boolean Algebras for Tasks and Value Functions

In this section, we develop the notion of a Boolean task algebra, allowing us to perform logical operations—conjunction ($\wedge$), disjunction ($\vee$) and negation ($\neg$)—over the space of tasks. We then show that, having solved a series of base tasks, an agent can use its knowledge to solve tasks expressible as a Boolean expression over those tasks, without any further learning.

We consider a family of related MDPs $\mathcal{M}$ restricted by the following assumptions:

**Assumption 1.** *For all tasks in a set of tasks $\mathcal{M}$, (i) the tasks share the same state space, action space and transition dynamics, (ii) the transition dynamics are deterministic, (iii) reward functions between tasks differ only on the absorbing set $\mathcal{G}$, and (iv) the set of possible terminal rewards consists of only two values. That is, for all $(g, a)$ in $\mathcal{G} \times \mathcal{A}$, we have that $r(g, a) \in \{r_\varnothing, r_\mathcal{U}\} \subset \mathbb{R}$ with $r_\varnothing \leq r_\mathcal{U}$. For all non-terminal states, we denote the reward $r_{s,a}$ to emphasise that it is constant across tasks.*

**Assumption 2.** *For all tasks in a set of tasks $\mathcal{M}$ which adhere to Assumption 1, the set of possible terminal rewards consists of only two values. That is, for all $(g, a)$ in $\mathcal{G} \times \mathcal{A}$, we have that $r(g, a) \in \{r_\varnothing, r_\mathcal{U}\} \subset \mathbb{R}$ with $r_\varnothing \leq r_\mathcal{U}$. For all non-terminal states, we denote the reward $r_{s,a}$ to emphasise that it is constant across tasks.*

Assumption 1 is similar to that of Todorov (2007) and identical to Van Niekerk et al. (2019), and imply that each task can be uniquely specified by its reward function. Furthermore, we note that Assumption 2 is only necessary to formally define the Boolean algebra. Although we have placed restrictions on the reward functions, the above formulation still allows for a large number of tasks to be represented. Importantly, sparse rewards can be formulated under these restrictions.

### 3.1    A Boolean Algebra for Tasks

An abstract Boolean algebra is a set $\mathcal{B}$ equipped with operators $\neg, \vee, \wedge$ that satisfy the Boolean axioms of (i) idempotence, (ii) commutativity, (iii) associativity, (iv) absorption, (v) distributivity, (vi) identity, and (vii) complements.[2] Given the above definitions and the restrictions placed on the set of tasks we consider, we can now define a Boolean algebra over a set of tasks.

**Theorem 1.** *Let $\mathcal{M}$ be a set of tasks. Define $\mathcal{M}_\mathcal{U}, \mathcal{M}_\varnothing \in \mathcal{M}$ to be tasks with the respective reward functions*

$$r_{\mathcal{M}_\mathcal{U}} : \mathcal{S} \times \mathcal{A} \to \mathbb{R} \qquad\qquad r_{\mathcal{M}_\varnothing} : \mathcal{S} \times \mathcal{A} \to \mathbb{R}$$

$$(s, a) \mapsto \begin{cases} r_\mathcal{U}, & \text{if } s \in \mathcal{G} \\ r_{s,a}, & \text{otherwise.} \end{cases} \qquad\qquad (s, a) \mapsto \begin{cases} r_\varnothing, & \text{if } s \in \mathcal{G} \\ r_{s,a}, & \text{otherwise.} \end{cases}$$

---

[1]Since we consider undiscounted MDPs, we can ensure the value function is bounded by augmenting the state space with a virtual state $\omega$ such that $\rho_{(s,a)}(\omega) = 1$ for all $(s, a)$ in $\mathcal{G} \times \mathcal{A}$, and $r = 0$ after reaching $\omega$.

[2]We provide a description of these axioms in the Appendix.

*Then $\mathcal{M}$ forms a Boolean algebra with universal bounds $\mathcal{M}_\varnothing$ and $\mathcal{M}_\mathcal{U}$ when equipped with the following operators:*

$$\neg : \mathcal{M} \to \mathcal{M}$$
$$M \mapsto (\mathcal{S}, \mathcal{A}, \rho, r_{\neg M}), \; where \; r_{\neg M} : \mathcal{S} \times \mathcal{A} \to \mathbb{R}$$
$$(s, a) \mapsto \left(r_{\mathcal{M}_\mathcal{U}}(s, a) + r_{\mathcal{M}_\varnothing}(s, a)\right) - r_M(s, a)$$

$$\vee : \mathcal{M} \times \mathcal{M} \to \mathcal{M}$$
$$(M_1, M_2) \mapsto (\mathcal{S}, \mathcal{A}, \rho, r_{M_1 \vee M_2}), \; where \; r_{M_1 \vee M_2} : \mathcal{S} \times \mathcal{A} \to \mathbb{R}$$
$$(s, a) \mapsto \max\{r_{M_1}(s, a), r_{M_2}(s, a)\}$$

$$\wedge : \mathcal{M} \times \mathcal{M} \to \mathcal{M}$$
$$(M_1, M_2) \mapsto (\mathcal{S}, \mathcal{A}, \rho, r_{M_1 \wedge M_2}), \; where \; r_{M_1 \wedge M_2} : \mathcal{S} \times \mathcal{A} \to \mathbb{R}$$
$$(s, a) \mapsto \min\{r_{M_1}(s, a), r_{M_2}(s, a)\}$$

*Proof.* See Appendix. □

Theorem 1 allows us to compose existing tasks together to create new tasks in a principled way. Figure 1 illustrates the semantics for each of the Boolean operators in a simple environment.

## 3.2 EXTENDED VALUE FUNCTIONS

The reward and value functions described in Section 2 are insufficient to solve tasks specified by the Boolean algebra above. We therefore extend these to define goal-oriented versions of the reward and value function, given by the following two definitions:

**Definition 1.** *The extended reward function $\bar{r} : \mathcal{S} \times \mathcal{G} \times \mathcal{A} \to \mathbb{R}$ is given by the mapping*

$$(s, g, a) \mapsto \begin{cases} N & if \; g \neq s \in \mathcal{G} \\ r(s, a) & otherwise, \end{cases} \tag{1}$$

*where $N \leq \min\{r_{MIN}, (r_{MIN} - r_{MAX})D\}$, and $D$ is the diameter of the MDP (Jaksch et al., 2010).*[3]

To understand why standard value functions are insufficient, consider two tasks that have multiple different goals, but at least one common goal. Clearly, there is a meaningful conjunction between them—namely, achieving the common goal. Now consider an agent that learns standard value functions for both tasks, and which is then required to solve their conjunction without further learning. Note that this is impossible in general, since the regular value function for each task only represents the value of each state with respect to the *nearest* goal. That is, for all states where the nearest goal for each task is *not* the common goal, the agent has no information about that common goal. Conversely, by learning extended value functions, the agent is able to learn the value of achieving all goals, and not simply the nearest one.

Because we require that tasks share the same transition dynamics, we also require that the absorbing set of states is shared. Thus the extended reward function adds the extra constraint that, if the agent enters a terminal state for a *different* task, it should receive the largest penalty possible. In practice, we can simply set $N$ to be the lowest finite value representable by the data type used for rewards.

**Definition 2.** *The extended Q-value function $\bar{Q} : \mathcal{S} \times \mathcal{G} \times \mathcal{A} \to \mathbb{R}$ is given by the mapping*

$$(s, g, a) \mapsto \bar{r}(s, g, a) + \int_{\mathcal{S}} \bar{V}^{\bar{\pi}}(s', g) \rho_{(s,a)}(ds'), \tag{2}$$

*where $\bar{V}^{\bar{\pi}}(s, g) = \mathbb{E}_{\bar{\pi}} \left[\sum_{t=0}^{\infty} \bar{r}(s_t, g, a_t)\right]$.* The extended Q-value function is similar to universal value function approximators (UVFAs) (Schaul et al., 2015), but differs in that it uses the extended reward function definition. It is also similar to DG functions (Kaelbling, 1993), except here we use task-dependent reward functions, as opposed to measuring distance between states.

The standard reward functions and value functions can be recovered from their extended versions through the following lemma.

---

[3]The diameter is defined as $D = \max_{s \neq s' \in \mathcal{S}} \min_{\pi} \mathbb{E}\left[T(s'|\pi, s)\right]$, where $T$ is the number of timesteps required to first reach $s'$ from $s$ under $\pi$.

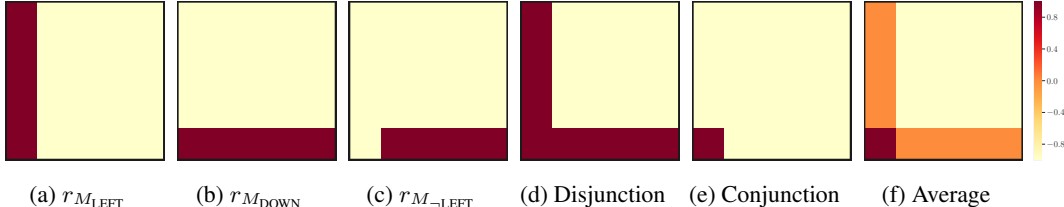

(a) $r_{M_{\text{LEFT}}}$  (b) $r_{M_{\text{DOWN}}}$  (c) $r_{M_{\neg\text{LEFT}}}$  (d) Disjunction  (e) Conjunction  (f) Average

Figure 1: Consider two tasks, $M_{\text{LEFT}}$ and $M_{\text{DOWN}}$, in which an agent must navigate to the left and bottom regions of an $xy$-plane respectively. From left to right we plot the reward for entering a region of the state space for the individual tasks, the negation of $M_{\text{LEFT}}$, and the union (disjunction) and intersection (conjunction) of tasks. For reference, we also plot the average reward function, which has been used in previous work to approximate the conjunction operator (Haarnoja et al., 2018; Hunt et al., 2019; Van Niekerk et al., 2019). Note that by averaging reward, terminal states that are not in the intersection are erroneously given rewards.

**Lemma 1.** *Let $r_M, \bar{r}_M, Q_M^*, \bar{Q}_M^*$ be the reward function, extended reward function, optimal Q-value function, and optimal extended Q-value function for a task $M$ in $\mathcal{M}$. Then for all $(s,a)$ in $\mathcal{S} \times \mathcal{A}$, we have (i) $r_M(s,a) = \max_{g \in \mathcal{G}} \bar{r}_M(s,g,a)$, and (ii) $Q_M^*(s,a) = \max_{g \in \mathcal{G}} \bar{Q}_M^*(s,g,a)$.*

*Proof.*

**(i):**

$$\max_{g \in \mathcal{G}} \bar{r}_M(s,g,a) = \begin{cases} \max\{N, r_M(s,a)\}, & \text{if } s \in \mathcal{G} \\ \max_{g \in \mathcal{G}} r_M(s,a), & \text{otherwise.} \end{cases}$$

$$= r_M(s,a) \qquad\qquad (N \le r_{\text{MIN}} \le r_M(s,a) \text{ by definition}).$$

**(ii):** Each $g$ in $\mathcal{G}$ can be thought of as defining an MDP $M_g := (\mathcal{S}, \mathcal{A}, \rho, r_{M_g})$ with reward function $r_{M_g}(s,a) := \bar{r}_M(s,g,a)$ and optimal Q-value function $Q_{M_g}^*(s,a) = \bar{Q}_M^*(s,g,a)$. Then using (i) we have $r_M(s,a) = \max_{g \in \mathcal{G}} r_{M_g}(s,a)$ and from Van Niekerk et al. (2019, Corollary 1), we have that $Q_M^*(s,a) = \max_{g \in \mathcal{G}} Q_{M_g}^*(s,a) = \max_{g \in \mathcal{G}} \bar{Q}_M^*(s,g,a)$.

$\square$

In the same way, we can also recover the optimal policy from these extended value functions by first applying Lemma 1, and acting greedily with respect to the resulting value function.

**Lemma 2.** *Denote $\mathcal{S}^- = \mathcal{S} \setminus \mathcal{G}$ as the non-terminal states of $\mathcal{M}$. Let $M_1, M_2 \in \mathcal{M}$, and let each $g$ in $\mathcal{G}$ define MDPs $M_{1,g}$ and $M_{2,g}$ with reward functions*

$$r_{M_{1,g}} := \bar{r}_{M_1}(s,g,a) \text{ and } r_{M_{2,g}} := \bar{r}_{M_2}(s,g,a) \text{ for all } (s,a) \text{ in } \mathcal{S} \times \mathcal{A}.$$

*Then for all $g$ in $\mathcal{G}$ and $s$ in $\mathcal{S}^-$,*

$$\pi_g^*(s) \in \arg\max_{a \in \mathcal{A}} Q_{M_{1,g}}^*(s,a) \text{ iff } \pi_g^*(s) \in \arg\max_{a \in \mathcal{A}} Q_{M_{2,g}}^*(s,a).$$

*Proof.* See Appendix.  $\square$

Combining Lemmas 1 and 2, we can extract the greedy action from the extended value function by first maximising over goals, and then selecting the maximising action: $\pi^*(s) \in \arg\max_{a \in \mathcal{A}} \max_{g \in \mathcal{G}} \bar{Q}^*(s,g,a)$. If we consider the extended value function to be a set of standard value functions (one for each goal), then this is equivalent to first performing generalised policy improvement (Barreto et al., 2017), and then selecting the greedy action.

Finally, much like the regular definition of value functions, the extended Q-value function can be written as the sum of rewards received by the agent until first encountering a terminal state.

**Corollary 1.** *Denote $G^*_{s:g,a}$ as the sum of rewards starting from $s$ and taking action $a$ up until, but not including, $g$. Then let $M \in \mathcal{M}$ and $\bar{Q}^*_M$ be the extended Q-value function. Then for all $s \in \mathcal{S}, g \in \mathcal{G}, a \in \mathcal{A}$, there exists a $G^*_{s:g,a} \in \mathbb{R}$ such that*

$$\bar{Q}^*_M(s, g, a) = G^*_{s:g,a} + \bar{r}_M(s', g, a'), \text{ where } s' \in \mathcal{G} \text{ and } a' = \arg\max_{b \in \mathcal{A}} \bar{r}_M(s', g, b).$$

*Proof.* This follows directly from Lemma 2. Since all tasks $M \in \mathcal{M}$ share the same optimal policy $\pi^*_g$ up to (but not including) the goal state $g \in \mathcal{G}$, their return $G^{\pi^*_g}_{T-1} = \sum_{t=0}^{T-1} r_M(s_t, \pi^*_g(s_t))$ is the same up to (but not including) $g$. $\qquad\square$

### 3.3 A Boolean Algebra for Value Functions

In the same manner we constructed a Boolean algebra over a set of tasks, we can also do so for a set of optimal extended Q-value functions for the corresponding tasks.

**Theorem 2.** *Let $\bar{\mathcal{Q}}^*$ be the set of optimal extended $\bar{Q}$-value functions for tasks in $\mathcal{M}$. Define $\bar{Q}^*_\varnothing, \bar{Q}^*_\mathcal{U} \in \bar{\mathcal{Q}}^*$ to be the optimal $\bar{Q}$-functions for the tasks $\mathcal{M}_\varnothing, \mathcal{M}_\mathcal{U} \in \mathcal{M}$. Then $\bar{\mathcal{Q}}^*$ forms a Boolean algebra when equipped with the following operators:*

$$\neg : \ \bar{\mathcal{Q}}^* \to \bar{\mathcal{Q}}^*$$
$$\bar{Q}^* \mapsto \neg\bar{Q}^*, \text{ where } \neg\bar{Q}^* : \ \mathcal{S} \times \mathcal{G} \times \mathcal{A} \to \mathbb{R}$$
$$(s, g, a) \mapsto \left(\bar{Q}^*_\mathcal{U}(s, g, a) + \bar{Q}^*_\varnothing(s, g, a)\right) - \bar{Q}^*(s, g, a)$$

$$\vee : \ \bar{\mathcal{Q}}^* \times \bar{\mathcal{Q}}^* \to \bar{\mathcal{Q}}^*$$
$$(\bar{Q}^*_1, \bar{Q}^*_2) \mapsto \bar{Q}^*_1 \vee \bar{Q}^*_2, \text{ where } \bar{Q}^*_1 \vee \bar{Q}^*_2 : \ \mathcal{S} \times \mathcal{G} \times \mathcal{A} \to \mathbb{R}$$
$$(s, g, a) \mapsto \max\{\bar{Q}^*_1(s, g, a), \bar{Q}^*_2(s, g, a)\}$$

$$\wedge : \ \bar{\mathcal{Q}}^* \times \bar{\mathcal{Q}}^* \to \bar{\mathcal{Q}}^*$$
$$(\bar{Q}^*_1, \bar{Q}^*_2) \mapsto \bar{Q}^*_1 \wedge \bar{Q}^*_2, \text{ where } \bar{Q}^*_1 \wedge \bar{Q}^*_2 : \ \mathcal{S} \times \mathcal{G} \times \mathcal{A} \to \mathbb{R}$$
$$(s, g, a) \mapsto \min\{\bar{Q}^*_1(s, g, a), \bar{Q}^*_2(s, g, a)\}$$

*Proof.* See Appendix. $\qquad\square$

### 3.4 Between Task and Value Function Algebras

Having established a Boolean algebra over tasks and extended value function, we finally show that there exists an equivalence between the two. As a result, if we can write down a task under the Boolean algebra, we can immediately write down the optimal value function for the task.

**Theorem 3.** *Let $\mathcal{F} : \mathcal{M} \to \bar{\mathcal{Q}}^*$ be any map from $\mathcal{M}$ to $\bar{\mathcal{Q}}^*$ such that $\mathcal{F}(M) = \bar{Q}^*_M$ for all $M$ in $\mathcal{M}$. Then $\mathcal{F}$ is a homomorphism.*

*Proof.* See Appendix. $\qquad\square$

## 4 Zero-shot Transfer Through Composition

We can use the theory developed in the previous sections to perform zero-shot transfer by first learning extended value functions for a set of base tasks, and then composing them to solve new tasks expressible under the Boolean algebra. To demonstrate this, we conduct a series of experiments in a Four Rooms domain (Sutton et al., 1999), where an agent must navigate in a grid world to a particular location. The agent can move in any of the four cardinal directions at each timestep, but colliding with a wall leaves the agent in the same location. The transition dynamics are deterministic, and rewards are $-0.1$ for all non-terminal states, and $1$ at the goal.

### 4.1 LEARNING BASE TASKS

We use a modified version of Q-learning (Watkins, 1989) to learn extended Q-value functions described previously. Our algorithm differs in a number of ways from standard Q-learning: we keep track of the set of terminating states seen so far, and at each timestep we update the extended Q-value function with respect to both the current state and action, as well as all goals encountered so far. We also use the definition of the extended reward function, and so if the agent encounters a terminal state of a different task, it receives reward $N$. The full pseudocode is listed in the Appendix.

If we know the set of goals (and hence potential base tasks) upfront, then it is easy to select a minimal set of base tasks that can be composed to produce the largest number of composite tasks. We first assign a Boolean label to each goal in a table, and then use the columns of the table as base tasks. The goals for each base task are then those goals with value 1 according to the table. In this domain, the two base tasks we select are $M_T$, which requires that the agent visit either of the top two rooms, and $M_L$, which requires visiting the two left rooms. We illustrate this selection procedure in the Appendix.

### 4.2 BOOLEAN COMPOSITION

Having learned the optimal extended value functions for our base tasks, we can now leverage Theorems 1–3 to solve new tasks with no further learning. Figure 2 illustrates this composition, where an agent is able to immediately solve complex tasks such as exclusive-or. We illustrate a few composite tasks here, but note that in general, if we have $K$ base tasks, then a Boolean algebra allows for $2^{2^K}$ new tasks to be constructed. Thus having trained on only two tasks, our agent has enough information to solve a total of 16 composite tasks.

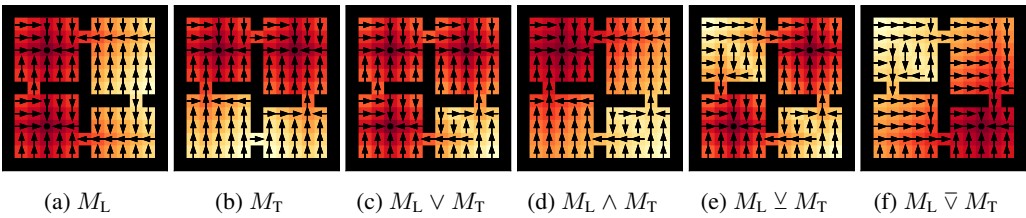

|     (a) $M_L$     |     (b) $M_T$     |     (c) $M_L \vee M_T$     |     (d) $M_L \wedge M_T$     |     (e) $M_L \veebar M_T$     |     (f) $M_L \barvee M_T$     |

Figure 2: An example of zero-shot Boolean algebraic composition using the learned extended value functions. Arrows represent the optimal action in a given state. (a–b) The learned optimal goal oriented value functions for the base tasks. (c) Zero-shot disjunctive composition. (d) Zero-shot conjunctive composition. (e) Combining operators to model exclusive-or composition. (f) Composition that produces logical nor. Note that the resulting optimal value function can attain a goal not explicitly represented by the base tasks.

By learning extended value functions, an agent can subsequently solve a massive number of tasks; however, the upfront cost of learning is likely to be higher. We investigate the trade-off between the two approaches by investigating how the sample complexity scales with the number of tasks. We compare to Van Niekerk et al. (2019), who used regular value functions to demonstrate optimal disjunctive composition. We note that while the upfront learning cost is therefore lower, the number of tasks expressible using only disjunction is $2^K - 1$, which is significantly less than the full Boolean algebra. We also test using an extended version of the Four Rooms domain, where additional goals are placed along the sides of all walls, resulting in a total of 40 goals. Empirical results are illustrated by Figure 3.

Our results show that while additional samples are needed to learn an extended value function, the agent is able to expand the tasks it can solve super-exponentially. Furthermore, the number of base tasks we need to solve is only logarithmic in the number of goal states. For an environment with $K$ goals, we need to learn only $\lfloor \log_2 K \rfloor + 1$ base tasks, as opposed to the disjunctive approach which requires $K$ base tasks. Thus by sacrificing sample efficiency initially, we achieve an exponential increase in abilities compared to previous work (Van Niekerk et al., 2019).

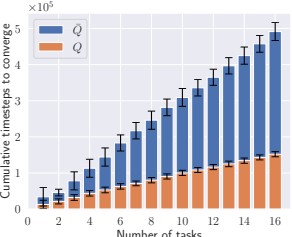
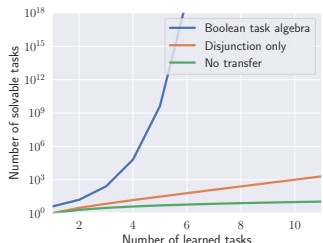
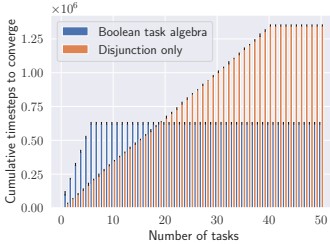

(a) Cumulative number of samples required to learn optimal extended and regular value functions. Error bars represent standard deviations over 100 runs.

(b) Number of tasks that can be solved as a function of the number of existing tasks solved. Results are plotted on a log-scale.

(c) Cumulative number of samples required to solve tasks in a 40-goal Four Rooms domain. Error bars represent standard deviations over 100 runs.

Figure 3: Results in comparison to the disjunctive composition of Van Niekerk et al. (2019). (a) The number of samples required to learn the extended value function is greater than learning a standard value function. However, both scale linearly and differ only by a constant factor. (b) The extended value functions allow us to solve exponentially more tasks than the disjunctive approach without further learning. (c) In the modified task with 40 goals, we need to learn only 7 base tasks, as opposed to 40 for the disjunctive case.

## 5 COMPOSITION WITH FUNCTION APPROXIMATION

Finally, we demonstrate that our compositional approach can also be used to tackle high-dimensional domains where function approximation is required. We use the same video game environment as Van Niekerk et al. (2019), where an agent must navigate a 2D world and collect objects of different shapes and colours. The state space is an $84 \times 84$ RGB image, and the agent is able to move in any of the four cardinal directions. The agent also possesses a `pick-up` action, which allows it to collect an object when standing on top of it. There are two shapes (squares and circles) and three colours (blue, beige and purple) for a total of six unique objects. The position of the agent is randomised at the start of each episode.

We modify deep Q-learning (Mnih et al., 2015) to learn extended action-value functions.[4] Our approach differs in that the network takes a goal state as additional input (again specified as an RGB image). Additionally, when a terminal state is encountered, it is added to the collection of goals seen so far, and when learning updates occur, these goals are sampled randomly from a replay buffer. We first learn to solve two base tasks: collecting blue objects, and collecting squares, which can then be composed to solve new tasks immediately.

We demonstrate composition characterised by (i) disjunction, (ii) conjunction and (iii) exclusive-or. This corresponds to tasks where the target items are: (i) blue or square, (ii) blue squares, and (iii) blue or squares, but not blue squares. Figure 4 illustrates sample trajectories, as well as the subsequent composed value functions, for the respective tasks.

## 6 RELATED WORK

The ability to compose value functions was first demonstrated using the linearly-solvable MDP framework (Todorov, 2007), where value functions could be composed to solve tasks similar to the disjunctive case (Todorov, 2009). Van Niekerk et al. (2019) show that the same kind of composition can be achieved using entropy-regularised RL (Fox et al., 2016), and extend the results to the standard RL setting, where agents can optimally solve the disjunctive case. Using entropy-regularised RL, Haarnoja et al. (2018) approximates the conjunction of tasks by averaging their reward functions, and demonstrates that by averaging the optimal value functions of the respective tasks, the agent can achieve performance close to optimal. Hunt et al. (2019) extends this result by composing value functions to solve the average reward task exactly, which approximates the true conjunctive case. More recently, Peng et al. (2019) introduce a few-shot learning approach to compose policies

---

[4]The hyperparameters and network architecture are listed in the Appendix

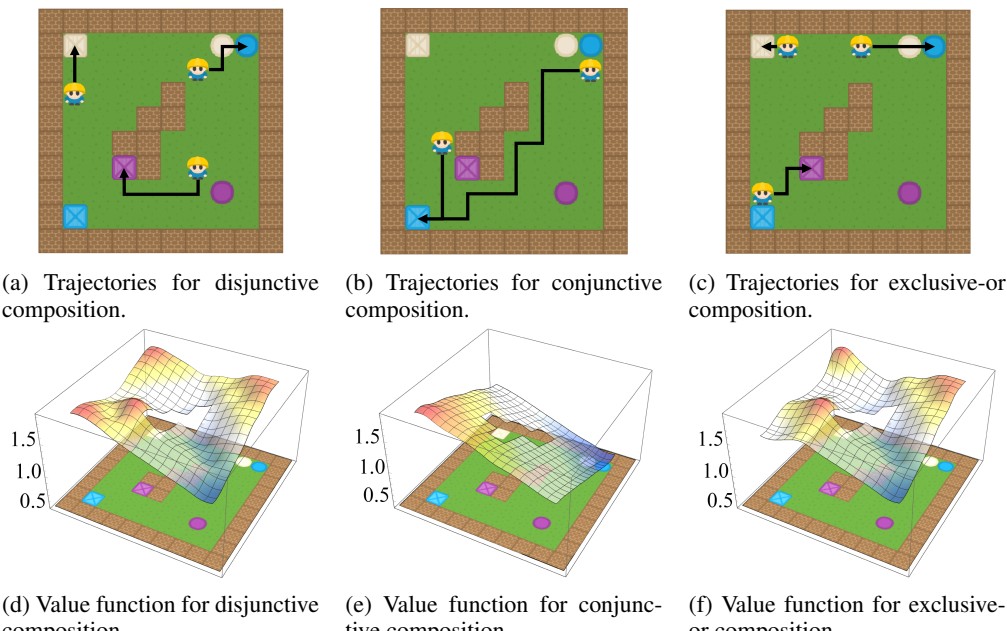

(a) Trajectories for disjunctive composition.

(b) Trajectories for conjunctive composition.

(c) Trajectories for exclusive-or composition.

(d) Value function for disjunctive composition.

(e) Value function for conjunctive composition.

(f) Value function for exclusive-or composition.

Figure 4: By composing extended value functions from the base tasks (collecting blue objects, and collecting squares), we can act optimally in new tasks with no further learning. To generate the value functions, we place the agent at every location and compute the maximum output of the network over all goals and actions. We then interpolate between the points to smooth the graph. Any error in the visualisation is due to the use of non-linear function approximation.

multiplicatively. Although lacking theoretical foundations, results show that an agent can learn a weighted composition of existing base skills to solve a new complex task. By contrast, we show that zero-shot optimal composition can be achieved for all Boolean operators.

# 7 CONCLUSION

We have shown how to compose tasks using the standard Boolean algebra operators. These composite tasks can be immediately solved by first learning goal-oriented value functions, and then composing them in a similar manner. Finally, we note that there is much room for improvement in learning the extended value functions for the base tasks. In our experiments, we learned each extended value function from scratch, but it is likely that having learned one for the first task, we could use it to initialise the extended value function for the second task to improve convergence times. One area for improvement lies in efficiently learning the extended value functions, as well as developing better algorithms for solving tasks with sparse rewards. For example, it is likely that approaches such as hindsight experience replay (Andrychowicz et al., 2017) could reduce the number of samples required to learn extended value functions, while Mirowski et al. (2017) provides a method for learning complex tasks with sparse rewards using auxiliary tasks. We leave incorporating these approaches to future work. Our proposed approach is a step towards both interpretable RL—since both the tasks and optimal value functions can be specified using Boolean operators—and the ultimate goal of lifelong learning agents, which are able to solve combinatorially many tasks in a sample-efficient manner.

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

## A  APPENDIX

### A.1  BOOLEAN ALGEBRA DEFINITION

**Definition 3.** *A Boolean algebra is a set $\mathcal{B}$ equipped with the binary operators $\vee$ (disjunction) and $\wedge$ (conjunction), and the unary operator $\neg$ (negation), which satisfies the following Boolean algebra axioms for $a, b, c$ in $\mathcal{B}$:*

  *(i)  Idempotence: $a \wedge a = a \vee a = a$.*

  *(ii)  Commutativity: $a \wedge b = b \wedge a$ and $a \vee b = b \vee a$.*

  *(iii)  Associativity: $a \wedge (b \wedge c) = (a \wedge b) \wedge c$ and $a \wedge (b \vee c) = (a \vee b) \vee c$.*

  *(iv)  Absorption: $a \wedge (a \vee b) = a \vee (a \wedge b) = a$.*

  *(v)  Distributivity: $a \wedge (b \vee c) = (a \wedge b) \vee (a \wedge c)$ and $a \vee (b \wedge c) = (a \vee b) \wedge (a \vee c)$.*

  *(vi)  Identity: there exists $\mathbf{0}, \mathbf{1}$ in $\mathcal{B}$ such that*

$$\mathbf{0} \wedge a = \mathbf{0}$$
$$\mathbf{0} \vee a = a$$
$$\mathbf{1} \wedge a = a$$
$$\mathbf{1} \vee a = \mathbf{1}$$

  *(vii)  Complements: for every $a$ in $\mathcal{B}$, there exists an element $a'$ in $\mathcal{B}$ such that $a \wedge a' = \mathbf{0}$ and $a \vee a' = \mathbf{1}$.*

### A.2  PROOF FOR THEOREM 1

**Theorem 1.** *Let $\mathcal{M}$ be a set of tasks. Define $\mathcal{M}_\mathcal{U}, \mathcal{M}_\varnothing \in \mathcal{M}$ to be tasks with the respective reward functions*

$$r_{\mathcal{M}_\mathcal{U}} : \mathcal{S} \times \mathcal{A} \to \mathbb{R} \qquad\qquad r_{\mathcal{M}_\varnothing} : \mathcal{S} \times \mathcal{A} \to \mathbb{R}$$

$$(s,a) \mapsto \begin{cases} r_\mathcal{U}, & \text{if } s \in \mathcal{G} \\ r_{s,a}, & \text{otherwise.} \end{cases} \qquad (s,a) \mapsto \begin{cases} r_\varnothing, & \text{if } s \in \mathcal{G} \\ r_{s,a}, & \text{otherwise.} \end{cases}$$

*Then $\mathcal{M}$ forms a Boolean algebra with universal bounds $\mathcal{M}_\varnothing$ and $\mathcal{M}_\mathcal{U}$ when equipped with the following operators:*

$$\neg : \mathcal{M} \to \mathcal{M}$$
$$M \mapsto (\mathcal{S}, \mathcal{A}, \rho, r_{\neg M}), \text{ where } r_{\neg M} : \mathcal{S} \times \mathcal{A} \to \mathbb{R}$$
$$(s,a) \mapsto \big( r_{\mathcal{M}_\mathcal{U}}(s,a) + r_{\mathcal{M}_\varnothing}(s,a) \big) - r_M(s,a)$$

$$\vee : \mathcal{M} \times \mathcal{M} \to \mathcal{M}$$
$$(M_1, M_2) \mapsto (\mathcal{S}, \mathcal{A}, \rho, r_{M_1 \vee M_2}), \text{ where } r_{M_1 \vee M_2} : \mathcal{S} \times \mathcal{A} \to \mathbb{R}$$
$$(s,a) \mapsto \max\{r_{M_1}(s,a), r_{M_2}(s,a)\}$$

$$\wedge : \mathcal{M} \times \mathcal{M} \to \mathcal{M}$$
$$(M_1, M_2) \mapsto (\mathcal{S}, \mathcal{A}, \rho, r_{M_1 \wedge M_2}), \text{ where } r_{M_1 \wedge M_2} : \mathcal{S} \times \mathcal{A} \to \mathbb{R}$$
$$(s,a) \mapsto \min\{r_{M_1}(s,a), r_{M_2}(s,a)\}$$

*Proof.* Let $M_1, M_2 \in \mathcal{M}$. We show that $\neg, \vee, \wedge$ satisfy the Boolean properties (i) – (vii).

**(i)–(v):** These easily follow from the fact that the $\min$ and $\max$ functions satisfy the idempotent, commutative, associative, absorption and distributive laws.

**(vi):** Let $r_{\mathcal{M}_\mathcal{U} \wedge M_1}$ and $r_{M_1}$ be the reward functions for $\mathcal{M}_\mathcal{U} \wedge M_1$ and $M_1$ respectively. Then for all $(s, a)$ in $\mathcal{S} \times \mathcal{A}$,

$$r_{\mathcal{M}_\mathcal{U} \wedge M_1}(s, a) = \begin{cases} \min\{r_\mathcal{U}, r_{M_1}(s, a)\}, & \text{if } s \in \mathcal{G} \\ \min\{r_{s,a}, r_{s,a}\}, & \text{otherwise.} \end{cases}$$

$$= \begin{cases} r_{M_1}(s, a), & \text{if } s \in \mathcal{G} \\ r_{s,a}, & \text{otherwise.} \end{cases} \qquad (r_{M_1}(s, a) \in \{r_\varnothing, r_\mathcal{U}\} \text{ for } s \in \mathcal{G})$$

$$= r_{M_1}(s, a).$$

Thus $\mathcal{M}_\mathcal{U} \wedge M_1 = M_1$. Similarly $\mathcal{M}_\mathcal{U} \vee M_1 = \mathcal{M}_\mathcal{U}$, $\mathcal{M}_\varnothing \wedge M_1 = \mathcal{M}_\varnothing$, and $\mathcal{M}_\varnothing \vee M_1 = M_1$. Hence $\mathcal{M}_\varnothing$ and $\mathcal{M}_\mathcal{U}$ are the universal bounds of $\mathcal{M}$.

**(vii):** Let $r_{M_1 \wedge \neg M_1}$ be the reward function for $M_1 \wedge \neg M_1$. Then for all $(s, a)$ in $\mathcal{S} \times \mathcal{A}$,

$$r_{M_1 \wedge \neg M_1}(s, a) = \begin{cases} \min\{r_{M_1}(s, a), (r_\mathcal{U} + r_\varnothing) - r_{M_1}(s, a)\}, & \text{if } s \in \mathcal{G} \\ \min\{r_{s,a}, (r_{s,a} + r_{s,a}) - r_{s,a}\}, & \text{otherwise.} \end{cases}$$

$$= \begin{cases} r_\varnothing, & \text{if } s \in \mathcal{G} \text{ and } r_{M_1}(s, a) = r_\mathcal{U} \\ r_\varnothing, & \text{if } s \in \mathcal{G} \text{ and } r_{M_1}(s, a) = r_\varnothing \\ r_{s,a}, & \text{otherwise.} \end{cases}$$

$$= r_{\mathcal{M}_\varnothing}(s, a).$$

Thus $M_1 \wedge \neg M_1 = \mathcal{M}_\varnothing$, and similarly $M_1 \vee \neg M_1 = \mathcal{M}_\mathcal{U}$.

$\square$

### A.3 PROOF FOR LEMMA 2

**Lemma 2.** *Denote $\mathcal{S}^- = \mathcal{S} \setminus \mathcal{G}$ as the non-terminal states of $\mathcal{M}$. Let $M_1, M_2 \in \mathcal{M}$, and let each $g$ in $\mathcal{G}$ define MDPs $M_{1,g}$ and $M_{2,g}$ with reward functions*

$$r_{M_{1,g}} := \bar{r}_{M_1}(s, g, a) \text{ and } r_{M_{2,g}} := \bar{r}_{M_2}(s, g, a) \text{ for all } (s, a) \text{ in } \mathcal{S} \times \mathcal{A}.$$

*Then for all $g$ in $\mathcal{G}$ and $s$ in $\mathcal{S}^-$,*

$$\pi_g^*(s) \in \arg\max_{a \in \mathcal{A}} Q_{M_{1,g}}^*(s, a) \text{ iff } \pi_g^*(s) \in \arg\max_{a \in \mathcal{A}} Q_{M_{2,g}}^*(s, a).$$

*Proof.* Let $g \in \mathcal{G}, s \in \mathcal{S}^-$ and let $\pi_g^*$ be defined by

$$\pi_g^*(s') \in \arg\max_{a \in \mathcal{A}} Q_{M_{1,g}}^*(s, a) \text{ for all } s' \in \mathcal{S}.$$

If $g$ is unreachable from $s$, then we are done since for all $(s', a)$ in $\mathcal{S} \times \mathcal{A}$ we have

$$g \neq s' \implies r_{M_{1,g}}(s', a) = \begin{cases} N, & \text{if } s' \in \mathcal{G} \\ r_{s',a}, & \text{otherwise} \end{cases} = r_{M_{2,g}}(s', a)$$

$$\implies M_{1,g} = M_{2,g}.$$

If $g$ *is* reachable from $s$, then we show that following $\pi_g^*$ must reach $g$. Since $\pi_g^*$ is proper, it must reach a terminal state $g' \in \mathcal{G}$. Assume $g' \neq g$. Let $\pi_g$ be a policy that produces the shortest trajectory

to $g$. Let $G^{\pi_g^*}$ and $G^{\pi_g}$ be the returns for the respective policies. Then,

$$G^{\pi_g^*} \geq G^{\pi_g}$$
$$\implies G^{\pi_g^*}_{T-1} + r_{M_{1,g}}(g', \pi_g^*(g')) \geq G^{\pi_g},$$

where $G^{\pi_g^*}_{T-1} = \sum_{t=0}^{T-1} r_{M_{1,g}}(s_t, \pi_g^*(s_t))$ and $T$ is the time at which $g'$ is reached.

$$\implies G^{\pi_g^*}_{T-1} + N \geq G^{\pi_g}, \text{ since } g \neq g' \in \mathcal{G}$$
$$\implies N \geq G^{\pi_g} - G^{\pi_g^*}_{T-1}$$
$$\implies (r_{\text{MIN}} - r_{\text{MAX}})D \geq G^{\pi_g} - G^{\pi_g^*}_{T-1}, \text{ by definition of } N$$
$$\implies G^{\pi_g^*}_{T-1} - r_{\text{MAX}}D \geq G^{\pi_g} - r_{\text{MIN}}D, \text{ since } G^{\pi_g} \geq r_{\text{MIN}}D$$
$$\implies G^{\pi_g^*}_{T-1} - r_{\text{MAX}}D \geq 0$$
$$\implies G^{\pi_g^*}_{T-1} \geq r_{\text{MAX}}D.$$

But this is a contradiction since the result obtained by following an optimal trajectory up to a terminal state without the reward for entering the terminal state must be strictly less that receiving $r_{\text{MAX}}$ for every step of the longest possible optimal trajectory. Hence we must have $g' = g$. Similarly, all optimal policies of $M_{2,g}$ must reach $g$. Hence $\pi_g^*(s) \in \arg\max_{a \in \mathcal{A}} Q^*_{M_{2,g}}(s, a)$. Since $M_1$ and $M_2$ are arbitrary elements of $\mathcal{M}$, the reverse implication holds too.

$\square$

### A.4 Proof for Theorem 2

**Theorem 2.** *Let $\bar{\mathcal{Q}}^*$ be the set of optimal extended $\bar{Q}$-value functions for tasks in $\mathcal{M}$. Define $\bar{Q}_\varnothing^*, \bar{Q}_\mathcal{U}^* \in \bar{\mathcal{Q}}^*$ to be the optimal $\bar{Q}$-functions for the tasks $\mathcal{M}_\varnothing, \mathcal{M}_\mathcal{U} \in \mathcal{M}$. Then $\bar{\mathcal{Q}}^*$ forms a Boolean algebra when equipped with the following operators:*

$$\neg : \bar{\mathcal{Q}}^* \to \bar{\mathcal{Q}}^*$$
$$\bar{Q}^* \mapsto \neg\bar{Q}^*, \text{ where } \neg\bar{Q}^* : \mathcal{S} \times \mathcal{G} \times \mathcal{A} \to \mathbb{R}$$
$$(s, g, a) \mapsto \left(\bar{Q}_\mathcal{U}^*(s, g, a) + \bar{Q}_\varnothing^*(s, g, a)\right) - \bar{Q}^*(s, g, a)$$

$$\vee : \bar{\mathcal{Q}}^* \times \bar{\mathcal{Q}}^* \to \bar{\mathcal{Q}}^*$$
$$(\bar{Q}_1^*, \bar{Q}_2^*) \mapsto \bar{Q}_1^* \vee \bar{Q}_2^*, \text{ where } \bar{Q}_1^* \vee \bar{Q}_2^* : \mathcal{S} \times \mathcal{G} \times \mathcal{A} \to \mathbb{R}$$
$$(s, g, a) \mapsto \max\{\bar{Q}_1^*(s, g, a), \bar{Q}_2^*(s, a)\}$$

$$\wedge : \bar{\mathcal{Q}}^* \times \bar{\mathcal{Q}}^* \to \bar{\mathcal{Q}}^*$$
$$(\bar{Q}_1^*, \bar{Q}_2^*) \mapsto \bar{Q}_1^* \wedge \bar{Q}_2^*, \text{ where } \bar{Q}_1^* \wedge \bar{Q}_2^* : \mathcal{S} \times \mathcal{G} \times \mathcal{A} \to \mathbb{R}$$
$$(s, g, a) \mapsto \min\{\bar{Q}_1^*(s, g, a), \bar{Q}_2^*(s, a)\}$$

*Proof.* Let $\bar{Q}^*_{M_1}, \bar{Q}^*_{M_2} \in \bar{\mathcal{Q}}^*$ be the optimal $\bar{Q}$-value functions for tasks $M_1, M_2 \in \mathcal{M}$ with reward functions $r_{M_1}$ and $r_{M_2}$. We show that $\neg, \vee, \wedge$ satisfy the Boolean properties (i) – (vii).

**(i)–(v):** These follow directly from the properties of the $\min$ and $\max$ functions.

**(vi):** For all $(s, g, a)$ in $\mathcal{S} \times \mathcal{G} \times \mathcal{A}$,

$$
\begin{aligned}
(\bar{Q}^*_{\mathcal{U}} \wedge \bar{Q}^*_{M_1})(s, g, a) &= \min\{(\bar{Q}^*_{\mathcal{U}}(s, g, a), \bar{Q}^*_{M_1}(s, g, a)\} \\
&= \min\{G^*_{s:g,a} + \bar{r}_{\mathcal{M}_{\mathcal{U}}}(s', g, a'), G^*_{s:g,a} + \bar{r}_{M_1}(s', g, a')\} \quad \text{(Corollary 1)} \\
&= G^*_{s:g,a} + \min\{\bar{r}_{\mathcal{M}_{\mathcal{U}}}(s', g, a'), \bar{r}_{M_1}(s', g, a')\} \\
&= G^*_{s:g,a} + \bar{r}_{M_1}(s', g, a') \qquad \text{(since } \bar{r}_{M_1}(s', g, a') \in \{r_\varnothing, r_{\mathcal{U}}, N\}) \\
&= \bar{Q}^*_{M_1}(s, g, a).
\end{aligned}
$$

Similarly, $\bar{Q}^*_{\mathcal{U}} \vee \bar{Q}^*_{M_1} = \bar{Q}^*_{\mathcal{U}}, \bar{Q}^*_{\varnothing} \wedge \bar{Q}^*_{M_1} = \bar{Q}^*_{\varnothing}$, and $\bar{Q}^*_{\varnothing} \vee \bar{Q}^*_{M_1} = \bar{Q}^*_{M_1}$.

**(vii):** For all $(s, g, a)$ in $\mathcal{S} \times \mathcal{G} \times \mathcal{A}$,

$$
\begin{aligned}
(\bar{Q}^*_{M_1} \wedge \neg\bar{Q}^*_{M_1})(s, g, a) &= \min\{\bar{Q}^*_{M_1}(s, g, a), (\bar{Q}^*_{\mathcal{U}}(s, g, a) - \bar{Q}^*_{\varnothing}(s, g, a)) - \bar{Q}^*_{M_1}(s, g, a)\} \\
&= G^*_{s:g,a} + \min\{\bar{r}_{M_1}(s', g, a'), (\bar{r}_{\mathcal{M}_{\mathcal{U}}}(s', g, a') + \bar{r}_{\mathcal{M}_{\varnothing}}(s', g, a')) \\
&\quad - \bar{r}_{M_1}(s', g, a')\} \\
&= G^*_{s:g,a} + \bar{r}_{\mathcal{M}_{\varnothing}}(s', g, a') \\
&= \bar{Q}^*_{\varnothing}(s, g, a).
\end{aligned}
$$

Similarly, $\bar{Q}^*_{M_1} \vee \neg\bar{Q}^*_{M_1} = \bar{Q}^*_{\mathcal{U}}$.

$\square$

## A.5 Proof for Theorem 3

**Theorem 3.** *Let $\mathcal{F} : \mathcal{M} \to \bar{\mathcal{Q}}^*$ be any map from $\mathcal{M}$ to $\bar{\mathcal{Q}}^*$ such that $\mathcal{F}(M) = \bar{Q}^*_M$ for all $M$ in $\mathcal{M}$. Then $\mathcal{F}$ is a homomorphism.*

*Proof.* Let $M_1, M_2 \in \mathcal{M}$. Then for all $(s, g, a)$ in $\mathcal{S} \times \mathcal{G} \times \mathcal{A}$,

$$
\begin{aligned}
\bar{Q}^*_{\neg M_1}(s, g, a) &= G^*_{s:g,a} + \bar{r}_{\neg M_1}(s', g, a') \quad \text{(from Corollary 1)} \\
&= G^*_{s:g,a} + (\bar{r}_{\mathcal{M}_{\mathcal{U}}}(s', g, a') + \bar{r}_{\mathcal{M}_{\varnothing}}(s', g, a')) - \bar{r}_{M_1}(s', g, a') \\
&= \left[(G^*_{s:g,a} + \bar{r}_{\mathcal{M}_{\mathcal{U}}}(s', g, a')) + (G^*_{s:g,a} + \bar{r}_{\mathcal{M}_{\varnothing}}(s', g, a'))\right] - (G^*_{s:g,a} + \bar{r}_{M_1}(s', g, a')) \\
&= \left[\bar{Q}^*_{\mathcal{U}}(s, g, a) + \bar{Q}^*_{\varnothing}(s, g, a)\right] - \bar{Q}^*_{M_1}(s, g, a) \\
&= \neg\bar{Q}^*_{M_1}(s, g, a) \\
\implies \mathcal{F}(\neg M_1) &= \neg\mathcal{F}(M_1)
\end{aligned}
$$

$$
\begin{aligned}
\bar{Q}^*_{M_1 \vee M_2}(s, g, a) &= G^*_{s:g,a} + \bar{r}_{M_1 \vee M_2}(s', g, a') \\
&= G^*_{s:g,a} + \max\{\bar{r}_{M_1}(s', g, a'), \bar{r}_{M_2}(s', g, a'')\} \\
&= \max\{G^*_{s:g,a} + \bar{r}_{M_1}(s', g, a'), G^*_{s:g,a} + \bar{r}_{M_2}(s', g, a'')\} \\
&= \max\{\bar{Q}^*_{M_1}(s, g, a), \bar{Q}^*_{M_2}(s, g, a)\} \\
&= (\bar{Q}^*_{M_1} \vee \bar{Q}^*_{M_2})(s, g, a) \\
\implies \mathcal{F}(M_1 \vee M_2) &= \mathcal{F}(M_1) \vee \mathcal{F}(M_2).
\end{aligned}
$$

Similarly $\mathcal{F}(M_1 \wedge M_2) = \mathcal{F}(M_1) \wedge \mathcal{F}(M_2)$.

$\square$

## A.6 GOAL-ORIENTED Q-LEARNING

Below we list the pseudocode for the modified Q-learning algorithm used in the four-rooms domain.

---

**Algorithm 1:** Goal-oriented $Q$-learning

---

**Input:** Learning rate $\alpha$, discount factor $\gamma$, exploration constant $\varepsilon$, lower-bound return $N$
Initialise $Q : \mathcal{S} \times \mathcal{S} \times \mathcal{A} \to \mathbb{R}$ arbitrarily
$\mathcal{G} \leftarrow \varnothing$
**while** *Q is not converged* **do**
    Initialise state $s$
    **while** *s is not terminal* **do**
        **if** $\mathcal{G} = \varnothing$ **then**
            Select random action $a$
        **else**

$$a \leftarrow \begin{cases} \arg\max\limits_{b \in \mathcal{A}} \left( \max\limits_{t \in \mathcal{G}} Q(s,t,b) \right) & \text{with probability } 1 - \varepsilon \\ \text{a random action} & \text{with probability } \varepsilon \end{cases}$$

        **end**
        Choose $a$ from $s$ according to policy derived from $Q$
        Take action $a$, observe $r$ and $s'$
        **foreach** $g \in \mathcal{G}$ **do**
            **if** $s'$ *is terminal* **then**
                **if** $s' \neq g$ **then**
                    $\delta \leftarrow N$
                **else**
                  $\delta \leftarrow r - Q(s,g,a)$
                **end**
            **else**
                $\delta \leftarrow r + \gamma \max_b Q(s',g,b) - Q(s,g,a)$
            **end**
            $Q(s,g,a) \leftarrow Q(s,g,a) + \alpha\delta$
        **end**
        $s \leftarrow s'$
    **end**
    $\mathcal{G} \leftarrow \mathcal{G} \cup \{s\}$
**end**
**return** $Q$

---

Figure 5: A $Q$-learning algorithm for learning extended value functions. Note that the greedy action selection step is equivalent to generalised policy improvement (Barreto et al., 2017) over the set of extended value functions.

## A.7 INVESTIGATING PRACTICAL CONSIDERATIONS

The theoretical results presented in this work rely on Assumptions 1 and 2, which restrict the tasks' transition dynamics and reward functions in potentially problematic ways. Although this is necessary to prove that Boolean algebraic composition results in optimal value functions, in this section we investigate whether these can be practically ignored. In particular, we investigate two restrictions: the requirement that tasks share the same terminal states, and the impact of using dense rewards.

### A.7.1 FOUR ROOMS EXPERIMENTS

We use the same setup as the experiment outlined in Section 4, but modify it in two ways. We first investigate the difference between using sparse and dense rewards. Our sparse reward function is defined as

$$r_{\text{sparse}}(s, a, s') = \begin{cases} 20 & \text{if } s' \in \mathcal{G} \\ -1 & \text{otherwise,} \end{cases}$$

and we use a dense reward function similar to Peng et al. (2019):

$$r_{\text{dense}}(s, a, s') = \frac{1}{|\mathcal{G}|} \sum_{g \in \mathcal{G}} \exp(\frac{|s' - g|^2}{4}) + r_{\text{sparse}}(s, a, s')$$

Using this dense reward function, we again learn to solve the two base task $M_T$ (reaching the centre of the top two rooms) and $M_L$ (reaching the centre of the left two rooms). We then compose them to solve a variety of tasks, with the resulting value functions illustrated by Figure 6.

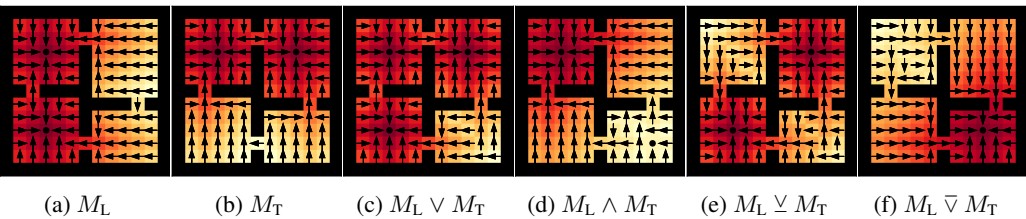

(a) $M_L$    (b) $M_T$    (c) $M_L \vee M_T$    (d) $M_L \wedge M_T$    (e) $M_L \veebar M_T$    (f) $M_L \barveebar M_T$

Figure 6: An example of Boolean algebraic composition using the learned extended value functions with dense rewards. Arrows represent the optimal action in a given state. (a–b) The learned optimal goal oriented value functions for the base tasks with dense rewards. (c) Disjunctive composition. (d) Conjunctive composition. (e) Combining operators to model exclusive-or composition. (f) Composition that produces logical nor. We note that the resulting value functions are very similar to those produced in the sparse reward setting.

We also modify the domain so that tasks need not share the same terminating states (that is, if the agent enters a terminating state for a *different* task, the episode does not terminate and the agent can continue as if it were a normal state). This results in four versions of the experiment:

  (i) `sparse reward, same absorbing set`
 (ii) `sparse reward, different absorbing set`
(iii) `dense reward, same absorbing set`
(iv) `dense reward, different absorbing set`

We learn extended value functions for each of the above setups, and then compose them to solve each of the $2^4$ tasks representable in the Boolean algebra. We measure each composed value functions by evaluating its policy in the sparse reward setting, averaging results over 100000 episodes. The results are given by Figure 7.

Our results indicate that extended value functions learned in the theoretically optimal manner (`sparse reward, same absorbing set`) are indeed optimal. However, for the majority of the tasks, relaxing the restrictions on terminal states and reward functions results in policies that are either identical or very close to optimal.

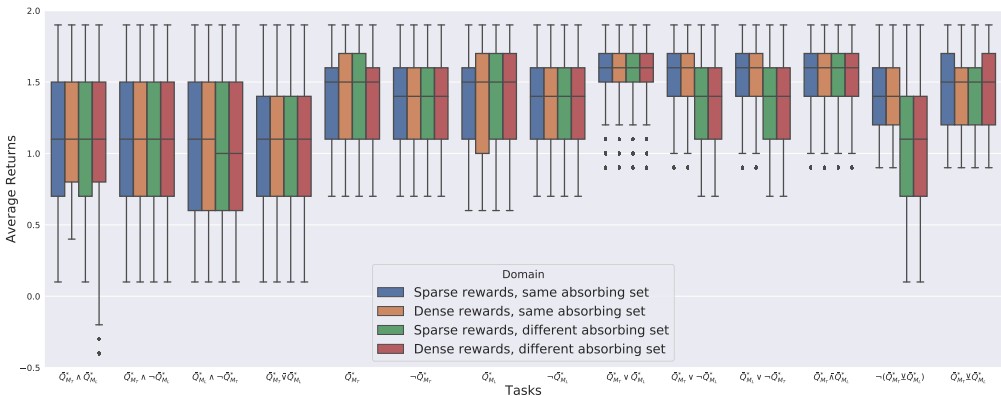

Figure 7: Box plots indicating returns for each of the 16 compositional tasks, and for each of the four variations of the domain. Results are collected over 100000 episodes with random start positions.

### A.7.2  FUNCTION APPROXIMATION EXPERIMENTS

In this section we investigate whether we can again loosen some of the restrictive assumptions when tackling high-dimensional environments. In particular, we run the same experiments as those presented in Section 5, but modify the domain so that (i) tasks need not share the same absorbing set, (ii) the `pickup-up` action is removed (the agent immediately collects an object when reaching it), and (iii) the position of every object is randomised at the start of each episode.

We first learn to solve three base tasks: collecting blue objects, collecting purple objects, and collecting squares , which can then be composed to solve new tasks immediately. We then demonstrate composition characterised by disjunction, conjunction and exclusive-or, with the resulting trajectories and value functions illustrated by Figure 8.

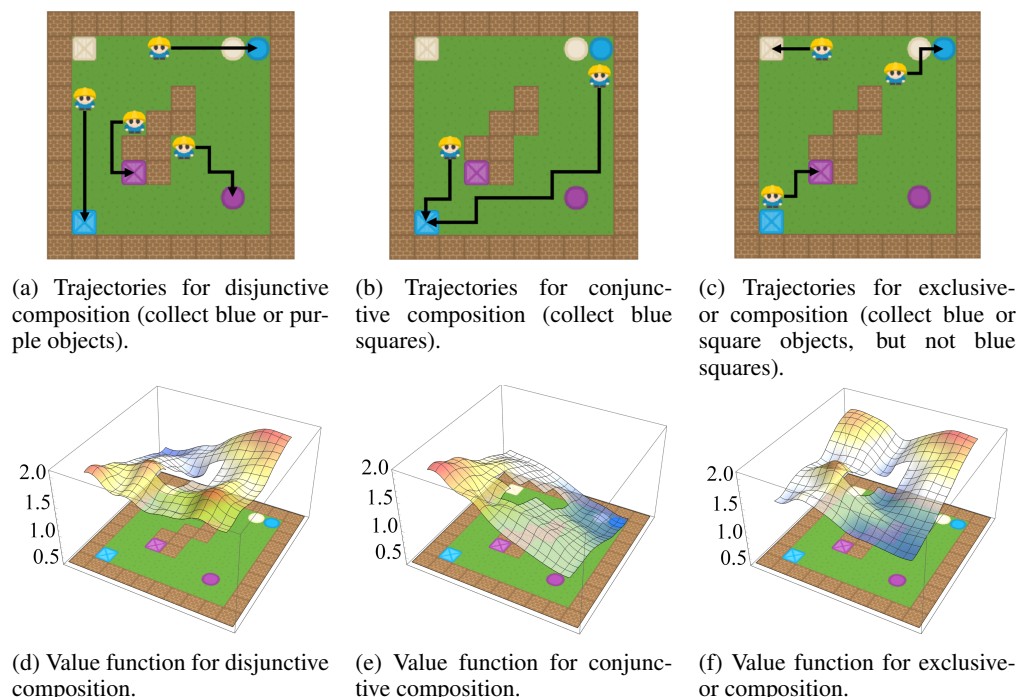

(a) Trajectories for disjunctive composition (collect blue or purple objects).

(b) Trajectories for conjunctive composition (collect blue squares).

(c) Trajectories for exclusive-or composition (collect blue or square objects, but not blue squares).

(d) Value function for disjunctive composition.

(e) Value function for conjunctive composition.

(f) Value function for exclusive-or composition.

Figure 8: Results for the video game environment with relaxed assumptions. We generate value functions to solve the disjunction of blue and purple tasks, and the conjunction and exclusive-or of blue and square tasks.

In summary, we have shown that our compositional approach offers strong empirical performance, even when the theoretical assumptions are violated. Finally, we expect that, in general, the errors due to these violations will be far outweighed by the errors due to non-linear function approximation.

## A.8 SELECTING BASE TASKS

The Four Rooms domain requires the agent to navigate to one of the centres of the rooms in the environment. Figure 9 illustrates the layout of the environment and the goals the agent must reach.

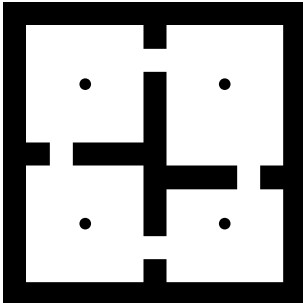

Figure 9: The layout of the Four Rooms domain. The circles indicate goals the agent must reach. We will refer to the goals as `top-left`, `top-right`, `bottom-left`, and `bottom-right`.

Since we know the goals upfront, we can select a minimal set of base tasks by assigning each goal a Boolean number, and then using the columns of the table to select the tasks. To illustrate, we assign Boolean numbers to the goals as follows:

| $x_1$ | $x_2$ | Goals |
|---|---|---|
| $r_\varnothing$ | $r_\varnothing$ | `bottom-right` |
| $r_\varnothing$ | $r_\mathcal{U}$ | `bottom-left` |
| $r_\mathcal{U}$ | $r_\varnothing$ | `top-right` |
| $r_\mathcal{U}$ | $r_\mathcal{U}$ | `top-left` |

Table 1: Assigning labels to the individual goals. The two Boolean variables, $x_1$ and $x_2$, represent the goals for the base tasks the agent will train on.

As there are four goals, we can represent each uniquely with just two Boolean variables. Each column in Table 1 represents a base task, where the set of goals for each task are those goals assigned a value $r_\mathcal{U}$. We thus have two base tasks corresponding to $x_1 = \{\texttt{top-right}, \texttt{top-left}\}$ and $x_2 = \{\texttt{bottom-left}, \texttt{top-left}\}$.

## A.9 DQN ARCHITECTURE AND HYPERPARAMETERS

In our experiments, we used a DQN with the following architecture:

1. Three convolutional layers:
    (a) Layer 1 has 6 input channels, 32 output channels, a kernel size of 8 and a stride of 4.
    (b) Layer 2 has 32 input channels, 64 output channels, a kernel size of 4 and a stride of 2.
    (c) Layer 3 has 64 input channels, 64 output channels, a kernel size of 3 and a stride of 1.
2. Two fully-connected linear layers:
    (a) Layer 1 has input size 3136 and output size 512 and uses a ReLU activation function.
    (b) Layer 2 has input size 512 and output size 4 with no activation function.

We used the ADAM optimiser with batch size 32 and a learning rate of $10^{-4}$. We trained every 4 timesteps and update the target Q-network every 1000 steps. Finally, we used $\epsilon$-greedy exploration, annealing $\epsilon$ to $0.01$ over $100000$ timesteps.

