# OpenReview forum: "A Boolean Task Algebra for Reinforcement Learning"
_ICLR.cc/2020/Conference — Reject_

### Official Review · AnonReviewer1 · 2019-10-22
**Official Blind Review #1**

**Rating:** 3

**Review:**

This paper proposes a new framework for defining Boolean algebra over the space of tasks in goal conditioned reinforcement learning and thereby achieving composition of tasks, defined by boolean operators, in zero-shot. The paper proves that with some assumptions made about a family of MDP’s, one can build Boolean algebra over the optimal Q-functions of the individual MDP and these Q-functions are equipped with all the mathematical operations that come with the Boolean algebra (e.g negation, conjunction). The paper verify their theoretical results by experiments in both the 4-room domain with standard Q-learning and in a simple video game domain with high-dimensional observation space and DQN. The proofs of all the theoretical results seem sound and the experiments support the theory. I enjoyed reading this paper as the paper is generally well written and the idea is quite neat.

That being said, I have a few concerns and questions about the paper that I would like the authors to respond to so I am leaning towards rejecting this paper at the moment. However, I will raise my score if the revision addresses my concerns or provide additional empirical evidence. My concerns are the following:

    1. My biggest concern is whether boolean algebra is the right abstraction/primitive for task level composition. Thus far, the most important application of boolean algebra has been in designing logic circuits where the individual components are quite simple. In the proposed framework, it seems that all of base tasks are required to be a well defined task which are already quite complex, so the utilities of composing them seems limited. For example, in the video game domain the author proposed, a very reasonable base task would be “collect white objects” -- this task when composed with the task “collect blue objects” is meaningless. This seems to be true for a large number of the MDP’s in the super-exponential composition. Furthermore, [1] also considers task level composition with sparse reward but I think these compositions cannot be expressed by boolean algebra. One of the most important appeal of RL is its generality so It would be great if the author can discuss the limitations of the proposed framework and provide an complex/real-world scenarios where composing these already complex base tasks are useful. Just writing would suffice as I understand setting up new environments can be difficult in short notice (Of course, actual experiments would be even better).

    2. Does the maze not change in the environment setup? (It would be nice if source code is provided) If that is the case I would like to see additional experiments on different mazes (i.e. different placement of walls and objects). In my opinion, if there is only a single maze, then the only thing that changes is the location of the agent which makes the task pretty easy and do not show the full benefit of function approximators. I think it’d strengthen the results if the framework generalizes to multiple and possibly unseen mazes.

    3. In the current formulation, a policy is discouraged to visit goals that are not in its current goal sets (receives lowest reward). While this could be just a proof artifact, it can have some performance implications. For example, in the 4 room domain, if I place a goal in the left corridor, then the agent in the bottom left room will need to take a longer route to reach top left (bottom left -> bottom right -> top right -> top left) instead of the shorter route (bottom left -> top left). From this perspective, it seems some non-trivial efforts need to be put into designing these "basis" tasks. I am curious about the discussion on this as well.

    4. Haarnoja et al. 2018 and other works on composing Q values can be applied to high-dimensional continuous control using actor-critic style algorithms and relies on the maximum entropy principle. Can the method proposed in this paper be used with actor-critic style? Is the max-entropy principle applicable here as well? Discussion would be great and experiments would be even better.

Out of all my concerns, 1 matters the most and I am willing to raise my score to weakly accept if it’s properly addressed. If, in addition, the authors could adequately address 2-4 I will raise my score to accept.

=======================================================================
Minor comments that did not affect my decision:
    - In definition 1, it would be nice to define r_min and r_max and g \ne s \in \mathcal{G} is also somewhat confusing.
    - In definition 2, \pi_g is never defined

Reference:
[1] Language as an Abstraction for Hierarchical Deep Reinforcement Learning, Jiang et al. 2019


**Experience Assessment:**

I have published one or two papers in this area.

**Review Assessment: Checking Correctness Of Derivations And Theory:**

I assessed the sensibility of the derivations and theory.

**Review Assessment: Checking Correctness Of Experiments:**

I carefully checked the experiments.

**Review Assessment: Thoroughness In Paper Reading:**

I read the paper thoroughly.

---

> ### Author Response · Authors · 2019-11-15
> **Reply to reviewer 1 (1/2)**
>
> Thank you for your careful review of our paper. We hope that the following points address your concerns.
>
> 1)
> >>> My biggest concern is whether boolean algebra is the right abstraction/primitive for task-level composition.
>
> - Note that Boolean algebra is the formal structure under which negation, disjunction, and conjunction operators are defined. Hence we use it because we are interested in formalising the negation, disjunction, and conjunction of tasks. While it is popularly associated with computer logics (logic circuits), it is actually also important to many other fundamental fields, notably set theory and propositional logics.
>
> - For some context/more intuition, consider the lifelong setting of a domestic robot. Say it has learned tasks like "make drink" (D), "make tea" (T), "make coffee" (C), “make sugary drink” (S), "make drink with milk" (M), etc. We then want to immediately be able to do,
>     - “make drink with sugar” : D AND S
>     - “make tea or any drink without coffee” : T OR (D AND NOT C)
>     - “make coffee or tea, with milk and without sugar" : (C OR T) AND (M AND NOT S)
>     - etc.
> - Note how all these statements seem like logical statements, even though there is no formal definition of logics over tasks. So intuitively we want to be able to,
>     - pose tasks as logical compositions of known tasks: For ease of task specification, rather than having to figure out the reward functions that will enable the agent to learn a desired composed task, and having to do it for every single desired composed task.
>     - immediately solve them: For ease of task completion, since learning tasks is hard. The more complex these tasks are and the more of them are desired, the more infeasible it becomes to learn all of them. For example, if there are say 1 billion achievable goals in an environment (as is easily the case in real life), then there are 2^(10^9) possible distinct tasks. An agent equipped with a Boolean algebra only needs to learn floor(log2(10^9))+1 = 30 base tasks to be able to solve any of that astronomical number of tasks.
>
> - The focus in this work is to achieve these intuitions formally by,
>     - Formally establishing logics over tasks: By formalising negation, disjunction, and conjunction of tasks under a Boolean Algebra.
>     - Formally showing zero-shot logical compositions of known tasks (i.e arbitrary negation, disjunction, and conjunction of tasks): By showing the homomorphism between the task and value function spaces.
>
>
> >>> For example, in the video game domain the author proposed, a very reasonable base task would be “collect white objects” -- this task when composed with the task “collect blue objects” is meaningless. This seems to be true for a large number of the MDP’s in the super-exponential composition.
>
> The conjunction of “collect white objects” and “ collect blue objects” is indeed meaningless, as it should be. But there is also meaningful compositions such as “collect objects that are not white”, “Collect objects that are not white and not blue”, etc. None of which would be possible formally without a Boolean algebra. Note that the meaningless composition “collect white objects that are blue” in the algebra reduces to “collect any objects with low desirability” (the lower universal bound task of the environment). Hence meaninglessness is also formally defined in the Boolean algebra, since it formalises logics over tasks.
>
> 2)
> >>> Does the maze not change in the environment setup?
>
> - The maze does not change. We have added another experiment where the agent and objects are randomly positioned at the start of each episode, and achieve similar results (see Appendix A.7.2).

---

> > ### Author Response · Authors · 2019-11-15
> > **Reply to reviewer 1 (2/2)**
> >
> >
> > 3)
> > >>> In the current formulation, a policy is discouraged to visit goals that are not in its current goal sets (receives lowest reward). While this could be just a proof artifact, it can have some performance implications. For example, in the 4 room domain, if I place a goal in the left corridor, then the agent in the bottom left room will need to take a longer route to reach top left (bottom left -> bottom right -> top right -> top left) instead of the shorter route (bottom left -> top left). From this perspective, it seems some non-trivial efforts need to be put into designing these "basis" tasks
> >
> > - The assumption that all tasks have the same transition dynamics is the reason why formally they also need to have the same absorbing states. We think of the absorbing set as the set of all achievable goals in the environment, and each task is simply defined by how desirable each of those goals are. The assumption that the reward functions only differ on the absorbing set ensures the agents experience before reaching goal states is consistent across all tasks.
> >
> > - If we want to adhere strictly to the theory, then in general, one can have an action that the agent chooses to achieve goals. For example, in the four-rooms experiments, we have a 5th action for “stay”, such that a goal position only becomes terminal if the agent chooses to stay in it. This represents the intuition that if an agent is at the goal location of a different task, and chooses to stay in it, then it has clearly chosen the wrong behaviour for the current task. Similarly, we have added a 5th action for “pickup” to the 2d game environment. The agent can now follow an optimal path to the goal objects, then choose to collect it. For more intuition, assume the agent is a garbage collector and the objects are garbage. Clearly, if we ask the agent to collect plastics to make their recycling easy, we do not want the robot to also collect other garbage objects.
> >
> > - All that being said, in practice we need not have the same absorbing set across all tasks (i.e the transition dynamics may differ in the absorbing sets). To demonstrate this, we have run additional experiments where we drop the constraint that the terminal states across tasks must be the same (see appendix A7).
> >
> >
> > 4)
> > >>> Can the method proposed in this paper be used with actor-critic style? Is the max-entropy principle applicable here as well? Discussion would be great and experiments would be even better.
> >
> > - Yes, since the homomorphism (Theorem 3) holds for any F, which is essentially a learning method. The extended value functions are goal oriented value functions and so the learning methods in multi-goal reinforcement learning are applicable to it. Hindsight Experience Replay [1] for example can be used to learn extended value functions in the sparse-rewards setting. Sophisticated learning methods can be employed to make learning extended value functions more efficient, but this is orthogonal to our main aim here - we leave this to future work. Finally, our results also make no assumption about the action space, and so readily extend to the continuous action setting.
> >
> > [1] Andrychowicz, Marcin, et al. "Hindsight experience replay." Advances in Neural Information Processing Systems. 2017.

---

> > > ### Comment · AnonReviewer1 · 2019-11-15
> > > **Response to rebuttal**
> > >
> > > Thank you for the explanation and new experimental results.
> > > 1) Regarding generality, I was more referring to compositions such as language. The example I gave is something like "Move the red ball to the left of green sphere" and I wonder if this kind of composition can be expressed through Boolean algebra. Intuitively, the colors/shapes can be swapped out and directions too. This kind of composition through structures like language which is ubiquitous.
> > > 2) Can you show some quantitative result and training curves or cummulated rewards? As it stands I can't tell how the performance is affected, and why did you remove the pick up action?
> > > 3) Thank you for the explanation.
> > > 4) I am more referring to how entropy will affect your value functions because it seems like your method strictly relies on the value function. Would entropy break some of the assumptions? Also zero-shot assumptions in this case becomes somewhere less appealing since you would have to retrain the actor every time.

---

> > > > ### Author Response · Authors · 2019-11-15
> > > > **Reply to reviewer 1**
> > > >
> > > > 2)
> > > > >>> Can you show some quantitative result and training curves or cummulated rewards?
> > > > - The boxplots showing average returns will be provided.
> > > >
> > > > >>> why did you remove the pick up action?
> > > > - The pickup action adds additional sparsity to the rewards in that the agent can only receive goal rewards if it chooses to pickup an object while on top of it. This makes training prohibitively difficult using standard DQN. This is why the main experiment had fixed object position (to reduce sparcity). In the different terminal states, random object positions setting, we are demonstrating that composition stills holds without the additional constraints. So adding more sparsity with the pickup action was not needed.
> > > > - However note that more sophisticated learning methods would solve this issue. We leave this to future work.
> > > >
> > > > 4)
> > > >
> > > > >>> I am more referring to how entropy will affect your value functions because it seems like your method strictly relies on the value function. Would entropy break some of the assumptions?
> > > > - In entropy-regularized RL the learned exponentiated values can used to recover the Q_values. So our method should still work in this setting.
> > > > - Since we make no assumption on the learning methods, other than that they should produce the extended value functions, we believe our method works with most learning methods.
> > > > - We leave proper investigation of these to future work.
> > > >
> > > > >>> Also zero-shot assumptions in this case becomes somewhere less appealing since you would have to retrain the actor every time.
> > > > - Note that our composition happens element wise, and the best action for a given goal remains un-changed even after composition (since composition of extended value functions produces extended value functions). All that changes during composition are the values per goal. Hence our composition method still works even with just the best value and actions for each state-goal.

---

### Official Review · AnonReviewer3 · 2019-10-23
**Official Blind Review #3**

**Rating:** 8

**Review:**

The paper proposes a method of combining value functions for a certain class of tasks, including shortest path problems, to solve composed tasks. By expressing tasks as a Boolean algebra, they can be combined using the negation, conjunction and disjunction operations. Analogous operations are available for the optimal value functions of the tasks, which allows the agent to have immediate access to the optimal policy of these composed tasks after solving the base tasks. The theoretical composition properties are confirmed empirically on the four rooms environment and with function approximation on a more complex domain.

The paper is generally well-written with a clear theoretical contribution and convincing experiments. The problem of composing tasks is important and I think this paper would be a good addition to the literature. My only concerns are in regards to the assumptions made in this formulation.
I would be wiling to increase my score if the authors address the following points:
1) Some further explanation on why extended value functions are necessary would be welcome at the beginning of section 3.2. Currently, it is only said that regular value functions are insufficient without any explanation. Also, additional motivation for the definition of extended value functions would be helpful to guide the reader.

2) Concerning assumption 1, it seems that the assumption that the reward functions only differ on the absorbing states is fairly limiting. For example, in a navigation task, if one goal location is A, then it must be an absorbing state under this formulation. So, if we have another goal location B, then we cannot use paths through A since it is set as absorbing, even though that A may be part of the shortest path to B. Would it be possible to modify this assumption to circumvent this problem?

3) In a similar vein, could the authors discuss possible limitations to this framework? For example, could this task/value composition be extended to arbitrary reward functions and continuing tasks or are there some fundamental limitations to this approach? If lifelong learning is a motivating setting for this work, it seems like dealing with non-episodic tasks and more complex rewards would be an important goal.

4) Fig. 3 b) does not seem to be particularly important as the result is clear enough in text. Perhaps the space could be used for something else.

As an aside, the paper is well-polished and the lack of typos is appreciated.


**Experience Assessment:**

I have published one or two papers in this area.

**Review Assessment: Checking Correctness Of Derivations And Theory:**

I assessed the sensibility of the derivations and theory.

**Review Assessment: Checking Correctness Of Experiments:**

I assessed the sensibility of the experiments.

**Review Assessment: Thoroughness In Paper Reading:**

I read the paper at least twice and used my best judgement in assessing the paper.

---

> ### Author Response · Authors · 2019-11-15
> **Reply to reviewer 3 (1/2)**
>
> Thank you for your careful review of our paper. We hope that the following points address your concerns.
>
> 1)
> >>> Additional motivation for the definition of extended value functions would be helpful to guide the reader.
>
> To understand why standard value functions are insufficient, consider two tasks that have multiple different goals, but at least one common goal. Clearly, there is a meaningful conjunction between them-namely, achieving the common goal. Now consider an agent that learns standard value functions for both tasks, and which is then required to solve their conjunction without further learning. Note that this is impossible in general, since the regular value function for each task only represents the value of each state with respect to the *nearest* goal. That is, for all states where the nearest goal for each task is *not* the common goal, the agent has no information about that common goal. Conversely, by learning extended value functions, the agent is able to learn the value of achieving all goals, and not simply the nearest one.
>
> >>> Some further explanation on why extended value functions are necessary would be welcome at the beginning of section 3.2
>
> As suggested we have added the above explanation in section 3.2.
>
>
> 2)
> >>> Concerning assumption 1, it seems that the assumption that the reward functions only differ on the absorbing states is fairly limiting. For example, in a navigation task, if one goal location is A, then it must be an absorbing state under this formulation.
>
> The assumption that all tasks have the same transition dynamics is the reason why formally they also need to have the same absorbing states. We think of the absorbing set as the set of all achievable goals in the environment, and each task is simply defined by how desirable each of those goals are. The assumption that the reward functions only differ on the absorbing set ensures the agents experience before reaching goal states is consistent across all tasks.
>
> >>> … If we have another goal location B, then we cannot use paths through A since it is set as absorbing, even though that A may be part of the shortest path to B.
>
> - If we want to adhere strictly to the theory, then in general, one can have an action that the agent chooses to achieve goals. For example, in the four-rooms experiments, we have a 5th action for “stay”, such that a goal position only becomes terminal if the agent chooses to stay in it. This represents the intuition that if an agent is at the goal location of a different task, and chooses to stay in it, then it has clearly chosen the wrong behaviour for the current task. Similarly, we have added a 5th action for “pickup” to the 2d game environment. The agent can now follow an optimal path to the goal objects, then choose to collect it. For more intuition, assume the agent is a garbage collector and the objects are garbage. Clearly, if we ask the agent to collect plastics to make their recycling easy, we do not want the robot to also collect other garbage objects.
> - All that being said, in practice we need not have the same absorbing set across all tasks (i.e the transition dynamics may differ in the absorbing sets). To demonstrate this, we have run additional experiments where we drop the constraint that the terminal states across tasks must be the same (see appendix A7), and achieve very similar results.
>
>
> 3)
> >>> Could this task/value composition be extended to arbitrary reward functions?
>
> - The assumptions we make are for theoretical rigour. Note that the only additional assumption in comparison to the literature [1] is assumption (iv), which is the assumption that ensures tasks have a Boolean nature so that the algebra can be formally established.
> - In practice our framework works even with dense rewards and different terminal states across task. We have added experiments in the four-rooms domain (see appendix A.7). Figure 7 shows the average return for all composed tasks after training the base tasks under various relaxations of our assumptions.
> - The key insight that enables this level of generality is the introduction of extended value functions.  While we provide a method for learning the extended value functions, a lot of work can still be done to make them more practical. The use of faster learning methods (such as hindsight experience replay) and better function approximators would improve learning these extended value functions. This is somewhat orthogonal to our main focus, but is certainly an important direction for future work.

---

> > ### Author Response · Authors · 2019-11-15
> > **Reply to reviewer 3 (2/2)**
> >
> > >>> Could this task/value composition be extended to continuing tasks?
> >
> > - In this work, we are interested in goal-reaching tasks, which are by nature episodic. There is disagreement about an exact definition lifelong learning, but here we take it to mean the setting where an agent is given tasks sampled from some distribution throughout its lifetime, and must solve each task in turn. This setting is formally defined in [2].
> > - Note that while the tasks are episodic, if needed an agent can simply be left to continually act in the environment. For example after learning how to collect blue objects, an agent can be left to continue collecting blue objects in the environment without terminating.
> >
> >
> > [1] Van Niekerk, Benjamin, et al. "Composing Value Functions in Reinforcement Learning." International Conference on Machine Learning. 2019.
> > [2] Abel, David, et al. "Policy and value transfer in lifelong reinforcement learning." International Conference on Machine Learning. 2018.

---

### Official Review · AnonReviewer2 · 2019-10-23
**Official Blind Review #2**

**Rating:** 3

**Review:**

This paper introduces a framework for composing tasks by treating tasks as a Boolean algebra. The paper assumes an undiscounted MDP with a 0-1 reward and a fixed absorbing set G, and considers a family of tasks defined by different reward functions. Each task defers only by the value of the reward function at the absorbing set G. These restrictions are quite severe but basically describes goal-state reaching sparse reward tasks, which are quite general and valuable to study. The paper then defines a mapping onto a Boolean algebra for these tasks and shows how the mapping also allows re-using optimal Q functions for each task to solve a Boolean composition of these tasks. This is demonstrated on the tabular four-rooms environment and using deep Q learning for a 2D navigation task.

The writing is relatively clear and the experiments support the claim in the paper that the framework allows learning compositions of skills. Both experiments show that after learning a set of base tasks, the method can solve a task in a zero-shot manner by composing Q functions according to the specified task. This capability seems very useful wherever it can be applied. But I worry that since the setting is so constrained, it is not likely to be widely applicable. The method in the paper likely does not apply to non-sparse, non-goal reaching settings, and prior methods have explored compositionality in that space anyways.

The coverage of prior work seems complete. One suggestion is to discuss recent goal relabeling work such as Hindsight Experience Replay (Andrychowicz 2017). Kaelbling 1993 is mentioned already, but this line of work has recently shown significant progress in learning to achieve multiple goals at the same time from a different perspective (and also considers sparse rewards).

However, my main concern with this paper is that it is not clear the language of Boolean algebra leads to significant insights in solving these compositional problems. Take Figure 1, which shows the disjunction and conjunction of tasks. While it is true the average does not lead to the same optimal policy as the conjunction, people use it because learning from the completely sparse reward is often prohibitively difficult. This kind of reasoning is straightforward in the restricted case of MDPs considered in the paper and people can design their reward function directly without considering boolean algebra. The result and proofs about recovering optimal Q functions without extra further training are interesting, but again, seem straightforward in the restricted family of MDPs considered without looking at Boolean algebra. Therefore, I am currently considering the paper borderline.


**Experience Assessment:**

I have read many papers in this area.

**Review Assessment: Checking Correctness Of Derivations And Theory:**

I assessed the sensibility of the derivations and theory.

**Review Assessment: Checking Correctness Of Experiments:**

I carefully checked the experiments.

**Review Assessment: Thoroughness In Paper Reading:**

I read the paper thoroughly.

---

> ### Author Response · Authors · 2019-11-15
> **Reply to reviewer 2 (1/2)**
>
> Thank you for your careful review of our paper. We hope that the following points address your concerns.
>
> 1)
> >>> I worry that since the setting is so constrained, it is not likely to be widely applicable. The method in the paper likely does not apply to non-sparse, non-goal reaching settings.
>
> - In view of your concern, we investigated the effect of dropping the constraints made on the reward functions (see appendix A.7). Figure 7 shows the average return for all composed tasks in the four-rooms domain after training the base tasks under various relaxations of our assumptions. It shows that our framework works even with dense rewards and different terminal states across task.
> - Note that goal reaching tasks and non-goal reaching tasks represent 2 different areas in reinforcement learning with large bodies of work. In this work, we are interested in goal-reaching tasks because they lend themselves to the lifelong setting where an agent is given tasks sampled from some distribution throughout its lifetime. This setting is formally defined in [1], and our work is a step towards achieving such lifelong agents.
> - Also with goal-reaching tasks, if needed an agent can simply be left to continually act in the environment even after achieving a goal. For example, after learning how to collect blue objects, an agent can be left to continue collecting blue objects in the environment without terminating [3].
>
>
>
> >>> ... prior methods have explored compositionality in that space anyways.
>
> - To the best of our knowledge, no prior methods in any reinforcement learning setting has explored optimal zero-shot composition of arbitrary negation, disjunction, and conjunction of tasks.
>
>
> 2)
> >>> One suggestion is to discuss recent goal relabeling work such as Hindsight Experience Replay (Andrychowicz 2017). Kaelbling 1993 is mentioned already, but this line of work has recently shown significant progress in learning to achieve multiple goals at the same time from a different perspective (and also considers sparse rewards).
>
> - Thank you for the reference to HER [2]. There has indeed been a lot of work on efficient learning in multi-goal RL, and these can be used to learn the extended value functions. HER for example can be used to learn extended value functions in the sparse-rewards setting.
> - Note however that this is orthogonal to our work, which is more focused on formalising logics over tasks and their zero-shot composition. However, important future work would indeed revolve around efficiently learning the base extended value functions, for which methods like HER would be relevant. We will add a discussion of this in the paper.

---

> > ### Author Response · Authors · 2019-11-15
> > **Reply to reviewer 2 (2/2)**
> >
> >
> > 3)
> > >>> it is not clear the language of Boolean algebra leads to significant insights in solving these compositional problems
> >
> > - Note that Boolean algebra is the formal structure under which negation, disjunction, and conjunction operators are defined. Hence we use the language of Boolean algebra because that's the language of negations, conjunctions, and disjunctions. While it is popularly associated with computer logics (logic circuits), it is actually also important to many other fundamental fields, notably set theory and propositional logics.
> >
> > - For some context/more intuition, consider learning the tasks “collect blue objects” (B) and “collect square objects” (S). We then want to immediately be able to do,
> >     - “collect blue objects or square objects” : B OR S
> >     - “collect blue squares” : B AND S
> >     - “collect square objects that are not blue” : S AND NOT B
> >     - “collect blue objects that are not squares” : B AND NOT S
> >     - “collect any objects that are not blue or squared” :  NOT (B OR S)
> >     - etc.
> > - Note how all these statements seem like logical statements, even though there is no formal definition of logics over tasks. So intuitively we want to be able to,
> >     - pose tasks as logical compositions of known tasks: For ease of task specification, rather than having to figure out the reward functions that will enable the agent to learn a desired composed task, and having to do it for every single desired composed task.
> >     - immediately solve them: For ease of task completion, since learning tasks is hard and the more we need to learn the more infeasible it becomes both in memory and time constraints. For example, if there are say 1 billion achievable goals in an environment (as is easily the case in real life), then there are 2^(10^9) possible distinct tasks. An agent equipped with a Boolean algebra only needs to learn floor(log2(10^9))+1 = 30 base tasks to be able to solve any of that astronomical number of tasks.
> >
> > - The focus in this work is to achieve these intuitions formally by,
> >     - Formally establishing logics over tasks: By formalising negation, disjunction, and conjunction of tasks under a Boolean Algebra.
> >     - Formally showing zero-shot logical compositions of known tasks (i.e arbitrary negation, disjunction, and conjunction of tasks): By showing the homomorphism between the task and value function spaces.
> >
> > - Meanwhile previous work focus on zero-shot conjunction (optimally by [3]) and disjunction (approximately by [4]), but none considers the general case of arbitrary negation, disjunction, and conjunction.
> >
> >
> > [1] Abel, David, et al. "Policy and value transfer in lifelong reinforcement learning." International Conference on Machine Learning. 2018.
> > [2] Andrychowicz, Marcin, et al. "Hindsight experience replay." Advances in Neural Information Processing Systems. 2017.
> > [3] Van Niekerk, Benjamin, et al. "Composing Value Functions in Reinforcement Learning." International Conference on Machine Learning. 2019.
> > [4] Haarnoja, Tuomas, et al. "Composable deep reinforcement learning for robotic manipulation." 2018 IEEE International Conference on Robotics and Automation (ICRA). IEEE, 2018.

---

> > > ### Comment · AnonReviewer2 · 2019-11-15
> > > **Response to rebuttal**
> > >
> > > Thank you for your detailed response and effort in running the new experiments.
> > >
> > > > To the best of our knowledge, no prior methods in any reinforcement learning setting has explored optimal zero-shot composition of arbitrary negation, disjunction, and conjunction of tasks.
> > >
> > > While this is true, I meant that references [3] and [4] you mention in your comment do address composing value functions for general reward functions, and the main reason that this method is able to handle negation, disjunction, and conjunction is because of the restricted sparse + goal reaching setting. So it is a more general method (with provably good composition) for a restricted set of MDPs. I really appreciate the extra experiments to show that the algorithm can be run on other MDPs, although much of the theoretical derivations do not apply in those cases.

---

> > > > ### Author Response · Authors · 2019-11-15
> > > > **Reply to reviewer 2**
> > > >
> > > > Thank you for your quick response, which is greatly appreciated. We would just like to add that in the updated version we have separated our assumptions into Assumption 1 and Assumption 2 to make it clear what assumptions are we adding to the literature and why they are necessary.
> > > >     - Assumption 1 is identical to that of [3].
> > > >     - Assumption 2 says for all tasks, goals are either desirable or not. It is introduced to give tasks the Boolean nature necessary for the Boolean algebra to be formalised. As for the intuition for this assumption, consider again the tasks "collect blue objects" (B) and "collect square objects" (S). What is the meaning of the composed task "collect blue objects that are not blue" (i.e B AND NOT B), or the composed task "collect square objects that are not squares" (i.e S AND NOT S)? Intuitively they are both equally meaningless. This is what the Boolean algebra formalises as the universal lower bounds of tasks, M_emptyset, which is defined as all goals are equally undesirable.
> > > >     - Our proofs for zero-shot negation, disjunction, and conjunction hold with just Assumption 1. This can be seen in the proof for the homomorphism (Theorem 3).
> > > >     - While zero-shot composition holds for the individual operators without Assumption 2, the homomorphism does not. This is because a homomorphism requires the operators to be defined in an algebraic structure, but without Assumption 2 that structure is lost.
> > > >
> > > > While zero-shot negation and conjunction without additional constraints is a contribution by itself, our work focuses on the more general logical compositions.
> > > > We hope to have motivated why this is necessary for lifelong agents, and why our additional assumption to achieve this is a necessary one.

---

### Decision · Program_Chairs · 2019-12-19

**Decision:**

Reject

**Comment:**

This paper considers the situation where a set of reinforcement learning tasks are related by means of a Boolean algebra.  The tasks considered are restricted to stochastic shortest path problems. The paper shows that learning goal-oriented value functions for subtasks enables the agent to solve new tasks (specified with boolean operations on the goal sets) in a zero-shot fashion.  Furthermore, the Boolean operations on tasks are transformed to simple arithmetic operations on the optimal action-value functions, enabling the zero short transfer to a new task to be computationally efficient. This approach to zero-shot transfer is tested in the four room domain without function approximation and a small video game with function approximation.

The reviewers found several strengths and weaknesses in the paper.  The paper was clearly written.  The experiments support the claim that the method supports zero-shot composition of goal-specified tasks.  The weaknesses lie in the restrictive assumptions.  These assumptions require deterministic transition dynamics, reward functions that only differ on the terminal absorbing states, and having only two different terminal reward values possible across all tasks.  These assumptions greatly restrict the applicability of the proposed method.  The author response and reviewer comments indicated that some aspects these restrictions can be softened in practice, but the form of composition described in this paper is restrictive.  The task restrictions also seem to limit the method's utility on general reinforcement learning problems.

The paper falls short of being ready for publication at ICLR.  Further justification of the restrictive assumptions is required to convince the readers that the forms of composition considered in this paper are adequately general.